# Future Options Redundancy Planning: Designing Multiple Pathways to Resilience in Urban and Landscape Systems Facing Complex Change

David J. Brunckhorst [1] and E. Jamie Trammell [2,*]

1    Geography and Environmental Science, University of Queensland, St Lucia 4067, Australia
2    Environmental Science, Policy, & Sustainability Program Southern Oregon University, Ashland, OR 97520, USA
*    Correspondence: trammelle@sou.edu; Tel.: +1-541-552-6496

**Abstract:** Urban systems include complex interactions and interdependencies with adjoining landscapes and regions. The pressures of change are complex, constant, and increasing. Declining biodiversity, ecosystem function, social institutions, and climate change underwrite serious sustainability challenges across urban, peri-urban, and 'natural' landscapes. Urban and other human 'development' often results in environmental damage that drives the need for ecological regeneration and restoration. Integration of interdisciplinary urban sciences and landscape sciences can guide the design of regenerative pathways and nature-positive sustainability. Social perceptions, however, tend to promote a cast-back view that favors the old 'locked-in' policy that attempts to restore 'what was' the former environment or ecosystem. Often, however, these are no longer suitable to the circumstances and future pressures of change. If urban design and planning disciplines are to help society anticipate change, we need to move from primarily deterministic approaches to those that probabilistically explore trajectories to future landscapes. Urban science and landscape design can now provide future regenerative capacity for resilient and continuous adaptation. Ongoing sustainability requires urban and landscape designs that provide ongoing anticipatory, restorative, nature-positive capacity in the context of future change and pressures. Complexity, connectivity, and redundancy are important system attributes of social-ecological systems creating adaptive capabilities. A diversity of plausible future social-ecological system responses provide several response options and redundancy, with multiple pathways to alternative sustainable futures, enhancing our adaptive capacity. A diversity of feasible responses increases the likelihood of sustaining ecological processes under changing conditions. We propose Future Options Redundancy (FOR) plans as a useful tool for nature-positive design. FOR plans are a variety of possible pathways and alternative futures defined using the characteristics of a social-ecological landscape context. Foresight design capabilities recognize in advance, the accumulating circumstances, along with policy and design opportunities for social-ecological system transformation options in urban-landscape spaces, that are nature-positive—the mark of a sustainable regenerative society.

**Keywords:** interdisciplinary urban sciences; landscape sciences; socio-ecological systems; alternative futures design; policy lock-in; nature-positive; sustainability policy; regenerative design

## 1. Introduction

The only certainty for the future is continuing change. In particular, rapidly developing urban landscapes reflect massive changes to their former, underlying natural ecological conditions. Nasseur [1] argued strongly for use of 'landscape' concepts as both a medium and method in analytical, experimental, and policy synthesis in urban ecological design. Urban contexts sit within local to global scales of social-ecological system interactions, in which humanity faces rapidly accelerating pressures from environmental degradation, disrupted ecosystem services, and subsequent increasing climate change impacts [2].

Various pressures of change create uncertainty, with cumulative and interdependent effects. In complex, interdependent social-ecological systems (SES), not only are the accumulated individual parts 'greater than the whole', but the interdependent responses of individuals can combine to produce infinite outcomes [3,4]. Referred to as emergent properties—that is, new conditions that develop out of interactions of interdependent social and ecological elements. The new conditions manifest themselves as uniquely different from the individual elements that created them [3]. Social-ecological system interactions include fast- and slow-moving variables, feedbacks, threshold effects, and reorganizing transformations [4,5]. Response conditions emanating from SES interactions are often at the heart of sustainability issues. Poor institutional responses to fast and slow variables, followed by rapid change (e.g., a flood, fire, storm), feeds back into social conditions and institutions, often further impacting timely policy innovation and response [6,7]. Poor, narrowly focused policy and planning, along with policy lock-in (e.g., antiquated 100-year floodplain delineation) and inadequate governance, contributes to human and nature-negative path dependences, increasing the likelihood of tipping points into unsustainable conditions and pathways with fewer response options to adapt to further changes [8].

The patterns of urban landscapes also reflect the internalization of social-ecological interactions driving change in species populations, natural resource capacity, ecosystem health, human settlements, land use, and institutions [9–11]. Responses to landscape change are reflected in policy and planning activity that, in turn, create new landscape change [12–14]. Thus, narrowly focused or 'command and control' solutions based on a reaction to landscape change by government policy rarely work [15,16]. Ecological restoration, rehabilitation, and rebuilding of functional biodiversity systems that can recoup or augment ecosystem service delivery are crucial for an ecologically positive sustainable future. Policy-focused, interdisciplinary knowledge systems are urgently required for societal adaptation towards equity and sustainability of human–nature interactions across urban and other landscape contexts [17,18].

Birkeland [2] points out the need for decision theory and policy to be much less narrow and bounded in ecologically negative ways, and that 'design' needs to open up creatively, using multiple dimensions, scales, and options towards ecologically positive and restorative social-ecological systems (see also [7,13]). Likewise, a less restricted adaptive design response and a reframing of ecological restoration are needed [19–22], but these must be better integrated into nature-positive policy and social transformations [3,14]. Sustainable and resilient SES will be restored through human interactions with nature [3]. However, the future will not be like the past. Given rapid landscape and climate change, it is questionable whether restoration to a past condition is appropriate, desirable, or even feasible [19,22]. Novel ecological systems could be more viable and desirable for ecosystem services and future adaptive capacity [21,23,24], including innovative design in built environments through emerging urban sciences [2,25].

An increasing number of urban and landscape scientists are urging an alternative future scenarios approach, using SES concepts, to help identify policy and social adaptations to accelerating pressures of change and environmental deterioration [3,6,26–29]. Hulse et al. [30] argue for a greater focus on examining probabilistic future change impacts or predicable 'surprises', SES design, and planning options for desirable (social-ecological) futures (see [31] for one example). Most recently, Quintero-Uribe et al. [32] noted the paucity of spatially explicit scenario approaches that explored the co-benefits of designs between multiple nature perspectives and conditions. Local community-based, participatory scenario planning, based on positive visions for urban systems, is one approach for guiding cities towards nature-positive sustainability [25,32].

This paper explores an approach using the design and testing of multiple scenarios, in SES context, that might be valuable in forging integrated restoration and transformation towards more sustainable futures. In light of current and future pressures of change, we consider that ecological restoration and social transformation need to be integrated and pursued together with explicit pathways towards adaptive, alternative landscape

futures. A geographical understanding of social-ecological interactions informs appropriate scales of community–environment context to apply alternative landscape futures scenario design and evaluation techniques [13,33]. Through understanding the appropriate scale of engagement, based on social-ecological interactions and past, present, and future pressures of change, new understandings of plausible alternative futures for restorative urban social-ecological systems can be pursued. Additionally, the development of foresight intelligence (the ability to regularly consider and evaluate the future) [34] to understand plausible future options can help our understanding of socio-ecological tipping points (SETPs). We suggest that creating a Future Options Redundancy (FOR) plan—a portfolio of possible pathways and alternative futures that honor the characteristics and capacity of a region—is essential for effective SES restoration and future sustainability in urban and other social-ecological contexts.

## 2. Social-Ecological Context and Engagement

### 2.1. Co-Management

Public engagement and clear social relevance are major challenges in the translation and implementation of sustainability sciences [15]. Socially and ecologically, context-relevant information must be synthesized into meaningful, appropriate, and applicable platforms for communicating clear paths of possible change and impacts [27,29,35,36]. For example, proposing ecological rehabilitation must also address opportunities for socio-cultural adoptability, economic sustainability, cooperation, and social equity [12,37,38]. Collaborative co-management, the sharing of responsibilities and power among government, local resource users, and other resident stakeholders, is important for effective resource governance and provides another opportunity for good public engagement in SES. Co-management can mobilize resources and knowledge across different management scales, can build institutional bridges, and leads to trust and coordination [39,40]. Thus, co-management is an important tool for social-ecological restoration and associated sustainability transformations.

### 2.2. Place-Based Engagement

Defining and protecting meaningful places and spaces are important goals in resource management, but they are equally important in urban and neighborhood settings. Understanding local systems and their interdependencies that spatially nest within larger contexts of landscape-scale systems allows us to see processes that materialize at broader scales, which cannot otherwise be seen at a local scale [1]. Identity and attachment to a neighborhood is often the driver for how local people invest in a given urban landscape and their place in it over time. Through the shaping of landscapes and urban settings, social-ecological system interactions create a sense of attachment and 'place' identity [1,33,41]. Such relationships generate opportunities to operationalize cross-scale interactions of property rights for resource use, official jurisdictions, and patterns of ecological processes [16,38]. Thus, 'place' is the geographical area where the primary local community of interest is embedded, where residents interrelate, develop networks of trust, and have an interest in their environment and local civic affairs [40,42]. Place-based contexts are critical for engagement in ecological stewardship and adaptive practices because they inherently integrate local ecological and social constituents. Such context also brings together resource uses and socio-economic dependencies, requiring local to regional social transformations (e.g., post mining, logging) to accompany ecological restoration. Indigenous communities often have a 'reciprocal restoration' relationship with their land and water resources that includes care and repair of ecosystems, with subsequent feedback reinforcing cultural relationships of adaptive ecological stewardship in that place [14,43]. Urban systems are particularly attractive for this type of place-based engagement [44].

*2.3. Eco-Civic Engagement*

To understand and geographically delineate spatial contexts for social and ecological restoration, three characteristics of SES are important. First, the landscape should reflect a range of similar biophysical and ecological characteristics [13,37,45]. Second, the landscape context should maximize the area that residents believe represents their communities of interest, attachment to place and identity, and ideally a sense civic engagement [33,42,45]. The third element relates to social-institutional nesting of vertical (institutions) and horizontal (geographically) related formal and informal networks of collaboration. Nested or polycentric governance delivers decision-making and actions at the lowest common denominator, but it does provide for the upscaling of representative decision-making to manage externalities [13,16]. In practice, nested governance can be used to design multi-scaled policies for dealing with externalities that come from conservation and resource use yet retain integration [39]. Techniques for 'eco-civic' regionalization help define contexts of nested spatial frameworks for sustainability planning and governance that integrate environment and neighborhood [13,33,37]. Spatially nested eco-civic frameworks contribute meaningful contexts, which is a relatable and relevant stage for multiple actors to collaborate towards sustainable alternative futures [41,42,46]. Nested spatial frameworks of SES have proven useful in the transformation of catchment management, local multi-tenure resource management, stream and wetland restoration [39,41], food security issues [47], conservation planning, land use planning [37,48], and examining spatially explicit landscape futures scenarios and climate change adaptation [26,27,36,49].

*2.4. Urban and Landscape Engagement*

Urban and landscape sciences provide valuable scaffolding to identify opportunities for positive change—a crucial construct for good governance and essential for generating policy-focused transformative pathways towards ecologically resilient futures [1]. Through understanding the multiple scales relevant to social-ecological systems, responses can be more effectively and amenably coordinated through co-management and knowledge building [39]. Understanding spatial contexts that reflect present to future scales of social-ecological interactions in urban settings would contribute much more effective and efficient anticipatory governance because it realigns the focus on multi-level policy design and decision-making that integrates knowledge about physical and social spaces, including the ecosystem function and changing conditions of those places [41].

**3. Restorative Social Ecology for Changing Urban Landscapes**

*3.1. Social Feedbacks*

While many challenging ecological degradation issues are obvious and pose serious threats to biodiversity, ecosystem services, economies, and social well-being, many are still debated or denied. Diverse stakeholders are likely to have conflicting opinions on some states of the environment. Negative influences of interrelated, slow-moving variables can be hard to see for both the public and policy makers [39]. The multilayered and enmeshed relationships of biodiversity, ecological function, and social systems influence production, delivery, and the supply and demand of ecosystem services, including long-term regenerative capacity [11,50]. The interactions of social-ecological systems simultaneously influence memory and reorganization amongst residents [9,10]. As well as conveying ecosystem services across urban landscapes, close interdependencies of system interactions can cast web-like networks transmitting accumulating negative impacts that are both visible and invisible [2,11,50,51]. These landscape externalities also have feedback loops over varying time lags that considerably disrupt social systems [4,22]. Some examples of social consequences include impacts on local economies, jobs, community cohesion, transport, and other dependent industries, in addition to ecosystem services such as air quality, water availability and quality, waste assimilation, food production, and nature buffering to storm, flood, and wildfire events.

### 3.2. Novel SES Communities

Relying on historical environmental conditions prior to urban development to guide ecological restoration and production of ecosystem services within and across neighborhoods or peri-urban settings is increasingly a problem [19,21,22,24]. Whilst ecosystem resilience is supported by ecological memory [9], pressures of urban change will increasingly push social-ecological systems across thresholds into new, re-combined elements or conditions of greater or lesser stability [5,20,21,24]. Climate change is now a considerable forcing pressure which will further increase the number of urban and other ecosystems referred to by Williams and Jackson [23] as "no-analogue communities", having novel combinations of species and ecosystem elements. Indeed, in urban and peri-urban contexts of changing climate and environmental conditions, communities will increasingly experience corresponding social reorganization with the advent of alternate and new states of "no-analogue" social-ecological conditions. Novel alternative designs of ecological and human assemblages need to be designed and assessed, not only in ecologically restorative interventions, but also in government strategies, policies, and plans to increase adaptive capacity [6,14,37]. Community residents and the general public, along with policy makers, must develop a clear understanding that continuing past decisions and actions create 'lock-in' trajectories [52] of unsustainable directions that are hard to escape, maladaptive, and offer no simple solution (Figure 1). Higgs and coworkers [22] emphasize the significant value of ecosystem function and processes over structure, and hence the multitude of possible trajectories for ecosystems over time. They argue that future restoration strategies should only use historical knowledge as a guide, not as a template. Higgs and coauthors emphasize the importance of considering multiple possible designs, pathways, and futures for ecosystems that will be more 'nature-positive' in a particular urban SES context, as opposed to simply "restoring" a past state of nature which can no longer perform biodiversity functions [22].

The complex and dynamic relationships active in coevolving landscapes suggests that new SES configurations, including urban environments, are emerging and will continue to emerge in the future [3,4,10] (Figure 1). As described earlier, landscape and urban sciences help us understand SES context and provide capacity to communicate preferred options and pathways for landscapes and regions. Policy focused, multi-attribute, social-ecological landscape modelling is becoming increasingly sophisticated and valuable in designing and testing alternative landscape futures and, consequently, in guiding policy and planning towards practical implementation [27,36,53]. Developing foresight capability, by understanding past change coupled with a novel design of plausible future change, can identify pathways and opportunities to alter the course towards alternative sustainable futures [7,34] (Figure 1).

### 3.3. Nature-Positive Options and Futures

Nature-positive future options should be sustained by diversity and redundancy. Redundancy refers to the capacity to substitute an alternative, to fill a gap or a worn-out condition. The substitute might contribute a similar role as the one replaced, or something different. Substituting a team player with slightly different skills or the ability to move a different way not only provides 'repair' to a gap but opens up other options for future plays. In this way, diverse SES have more resilience to perturbations and more capacity to move in (nature-positive) directions without system collapse [3,6,39]. The recombinant and self-organizing capacities of diverse systems provide the functional means for redundancy to play its role in maintaining ecosystem functions and processes [21,24]. This principle is highly applicable to urban systems, comparable to financial portfolios in which diversification and the maintenance of complexity across assets and investments contributes better risk management by delivering stable returns over longer time frames despite uncertainty, perturbations, or flux in the performance of individual assets [6]. Therefore, diversity and redundancy are interrelated and work together to contribute nature-positive futures, and they are particularly important when systems are under

various negative pressures of change. Ecological restoration, coupled with restorative social and institutional transformation, must incorporate biological and ecosystem diversity that enhances the delivery mechanisms of ecosystem services with both resilient and adaptive capacities. Institutional redundancy contributing resilience simply requires thinking ahead about what might happen and what suite of policy pathways might be beneficial to pursue in various circumstances. It avoids policy lock-in, 'surprise', and reactive command-control by anticipating changing circumstances and by developing diverse policy and strategy responses and pathways to more positive outcomes and subsequent futures [3,6,7,14,31].

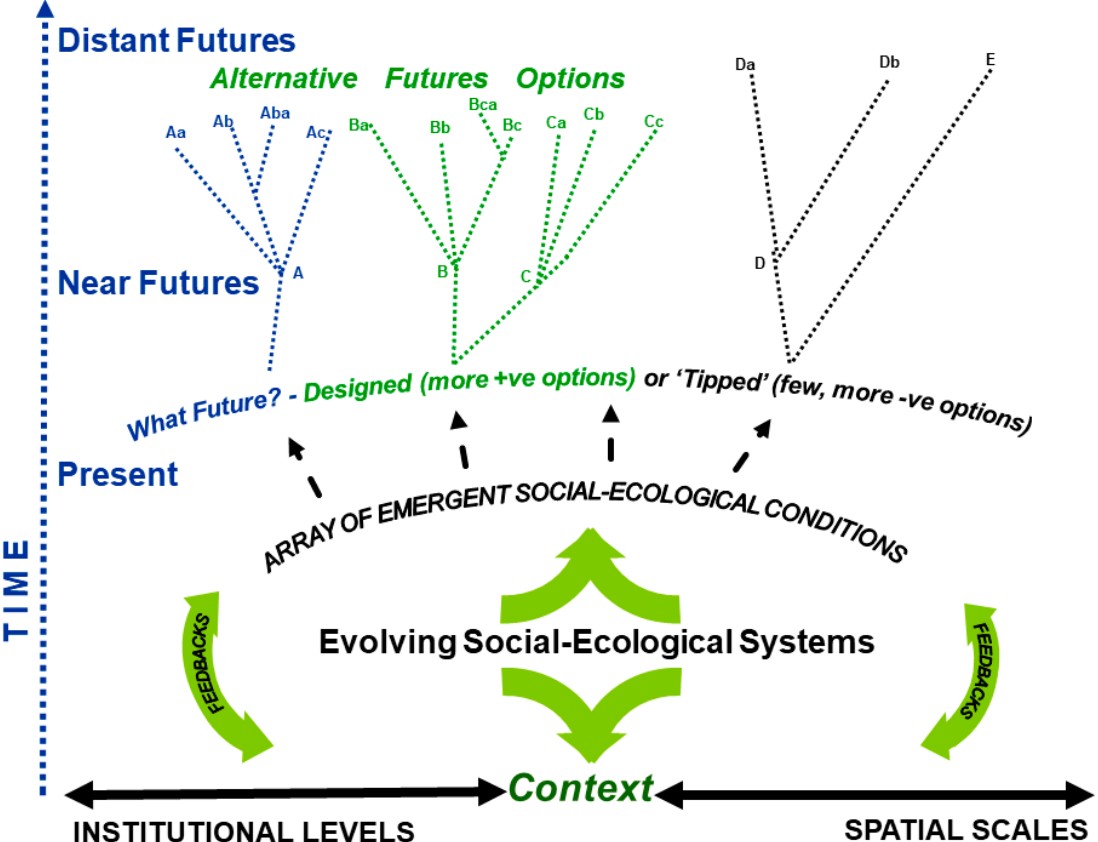

**Figure 1.** To contribute to nature-positive future design and planning, urban and landscape sciences need to understand multiple scales of social-ecological systems interactions that create context and re-organizing potential for adaptive options and pathways to near and distant futures (e.g., green in figure). However, narrowly focused policy and planning along with policy lock-in and inadequate governance contributes to negative path dependences and increasing impacts, increasing the likelihood of 'tipping points' into unsustainable pathways with fewer response options (e.g., black in figure). A diversity of plausible responses, instead of a single possible future (e.g., blue in figure), increases the likelihood of facilitating pathways to sustain ecological processes under emerging (near-term) and longer-term changing conditions. Future Options Redundancy (FOR) plans constitute a range of plausible pathways and alternative futures designs (e.g., green in figure) within the characteristics of a particular social-ecological context. FOR plans provide a place-based framework for designing nature-positive alternatives to future changes. (Adapted after [7].)

Dunwiddle and colleagues [54] discussed three types of ecological redundancy, which we consider to be equally important in urban systems design. Firstly, component redundancy, which is increased numbers of species and community complexity. Secondly, functional redundancy, which refers to substituting ecologically equivalent species, providing a recombined ecosystem which has species groups that can undertake similar functions. Thirdly, greater connectivity, which increases options for nature to take its course unimpeded. Elements of connectivity include increased numbers of linkages or connections, size

of connections for easy movement, proximity (suitable habitats or 'sources' within easy reach), and greater complexity of interconnections and networks [55,56].

Such landscape ecological attributes are just as relevant to urban, peri-urban, and rural social systems and their respective neighborhoods and communities [1,25,32]. Connectivity in social networks of communities, for example, and redundancy in human resource capital and collective knowledge of communities, contribute resilience through alternative options to dealing with climate change events, such as floods and storms. Thus, SES research needs to be more integrated with different social sciences, such as psychology, sociology, economics, and policy analysis, in order to design and evaluate adaptive future options [56].

## 4. Restorative Futures for Social-Ecological Systems

### 4.1. SES Feedbacks

In urban and peri-urban contexts, restoring ecological functions and ecosystem services along with their social systems requires a bigger vision that is more holistically integrated and includes implementation pathways. Analyses of alternative future scenarios for regional landscapes confronted with multiple pressures and uncertainties can provide valuable insights for social-ecological restoration and future resilience [36,48]. For example, the reduction in use of coal for heating and electricity generation, and the subsequent closure of coalmines that will require ecosystem rehabilitation, are not isolated from other change pressures in a particular region. Other changes and uncertainties in employment, education, transport and service infrastructure, ports and railways, urban growth or senescence, aging population, changing skills and employment, potable water availability, and so on will require careful attention when considering these scenarios. In this case, the big picture for social-ecological restoration and policy transformation will need a holistic examination of a range of alternative social-ecological reconfigurations to identify the variety of plausible futures, and an assessment of the perceived benefits and drawbacks of the impacts of each future scenario [8,27,35,47]. The restoration required for SES transformation embodies so much more than the mine site. With geographical knowledge of the SES context, forward planning options, including potential pathways for nature-positive change, can lead to a desirable set of futures.

While social-ecological systems possess self-organizing abilities and potential for recombination's that are nature-positive, SES connectivity, complexity, and redundancy are also necessary system attributes enabling adaptive capabilities to develop and function. Having a diversity of possible responses contributes capacity to sustain ecological processes experiencing changing conditions [6]. A portfolio of readily available social-ecological system responses contributes to and allows for recombination and redundancy. We call this 'Future Options Redundancy' (FOR), which is the array of possible pathways and alternative futures appropriate for the characteristics and capacity of that particular geographical region. FOR is a necessary component for the timely adaptation to change pressures. Elements of social redundancy, such as human resources capacity (e.g., recombining skills and training), wisdom (through community understanding of feedbacks and future options) of the social-institutional system in a particular context, and emerging circumstances (social, political, etc.), are also important to pursue sustainability transformations [7]. Coupling the understanding of opportunities for SES transformation with FOR plans could facilitate shifts, across transition thresholds, towards preferred, ecologically sustainable future conditions [6,20,21,24,28]. Public policy-focused interdisciplinary sciences can inform society of the preferred options and pathways towards long-term, nature-positive futures, as well as help recognize the most desirable and plausible moment when social conditions provide the opportunity to transform [6,7,57]. Similar to how designed redundancy in critical infrastructure (water, electricity, internet) is required in most urban landscapes, planning and policy makers can design redundancy in the landscape that provides multiple desirable outcomes, despite possible perturbations.

Social-ecological relationships are like ecosystem relationships, constantly changing and shifting, which creates uncertainty. Despite this uncertainty, we need to be ready to act

by well-informed responses, to move down the novel pathways of transformation [6,34,50]. Transformation risks will be minimized by integrating FOR planning with anticipatory action that maximizes the use of interdisciplinary data integration and optimized through monitoring, feedback, connectivity, and policy flexibility [6,14,15,28,41,58]. Even when the benefits of an anticipated pathway and action are clear, it is often difficult to push through the (often social) resistance, especially towards nature-positive transformation. Novel approaches, with a previously discussed flexibility, will be required, as will anticipatory procedures and social facilitation, in having future options which are ready for implementation when the context and circumstances shift to a window of opportunity that makes the SES shift achievable [3,28,41,59].

### 4.2. Alternative Landscape Futures

Utilizing a landscape modus allows for spatial integration of ecological, economic, and social variables to understand the spatial dimensions of change and impacts [60]. Analytically based future scenarios can build consensus on the kind of sustainable future we want [6,18,41,61]. Long-term horizon, preemptive, spatial analyses intend to inform governments and stakeholders of the impacts of different options for future land and other resource allocations. Spatially attributed visualization of likely emergent patterns and processes on landscapes and regions are further benefits of such integrated approaches for understanding the social-ecological interactions and landscape futures [29]. Alternative Landscape Futures (ALF) analysis contributes anticipatory knowledge through a geographic synthesis of the multiple policies, plans, and regulations that reflect existing and potential trajectories of change that influence the long-term resilience of social-ecological systems. Landscape futures analysis contributes FOR knowledge and FOR plans by integrating spatially explicit interdisciplinary data of trending trajectories of regional and evaluated designs of alternative landscape scenarios [35,36,62,63]. ALF scenario analysis elucidates future impact and risk, such as climate change-exacerbated flood and storm surge [27,41], as well as hazards and vulnerabilities from land use and development-related uncertainties that negatively impact ecosystem services, human services, safety, and well-being [26,36,48,55]. Scenario planning provides capacity to assess circumstances with varying uncertainty and connectivity between elements and interrelated impacts, and it can assist in the identification of likely emergent conditions [27,36,53,59]. FOR knowledge and plans will help guide decision choices of stakeholders and governments faced with policy, planning, and pathway alternatives, helping to facilitate more nature-positive adaptation and transformation. Citizen and expert workshop discussions of change pressures, trends, and uncertainties, conducted in a way that is both focused and contextually relevant, are useful starting points to understand current trajectories for ALF design and analysis [30,31,36,63]. Spatial derivation and mapping of trend futures and alternative futures provides a means to compare and contrast the impacts and benefits of multiple plausible, future landscapes, allowing a proactive assessment of sustainability. In this way, planning and policy making, through the design of multiple alternative futures, can increase redundancy.

### 4.3. SES Foresight

Poor environmental policy and planning decisions often stem from 'command and control' government. Policy 'lock-in' further reduces adaptive capacity. Inflexible policy and plans assume stability and linear causality. It is maladaptive, only reacting to impactful change when it happens—"Oh dear, not another unexpected natural disaster!" An informed strategically adaptive policy approach assumes change and manages for uncertainty by building a variety of adaptive capacity, including through strategically integrated SES landscape analysis grounded in landscape and urban sciences. This approach reduces political distortion and 'lock-in' by focusing policy makers on what they have previously failed to learn from the interdependencies of social-ecological systems [7,30]. Schindler and Hilborn [6] noted that ecological management will always operate with uncertainty and

recommended that policy deliberation be informed through examining alternative futures scenarios and embracing flexibility: "The best management and conservation plans will likely be those that can harness unexpected opportunities" [6] (p. 954).

The science–policy nexus itself needs a transformative, 'futures-focused' renewal of approaches in research and practice. Combined with new policy and institutional analytical methods to recognize and harness opportunities for transformation, landscape and urban science can further contribute to nature-positive policy and planning of alternative futures in ways that are flexible, adaptable, and implementable [7]. Advancing the understanding of multiple, nature-positive design pathways, along with capacities to identify and facilitate required policy processes towards alternative and ecologically resilient futures, would contribute restorative, nature-positive redundancy options under differing circumstances of complex change pressures.

## 5. Summary and Conclusions

Interdependencies of complexly intertwined social and ecological systems affect how change manifests on a landscape, which in turn shapes future ecological adaptive capacity for long-term sustainability. Policy and planning efforts to overcome declining ecosystem function and biodiversity rarely focus on understanding the emergent properties and self-organizing capacities of social-ecological system interactions that permit (multiple) future adaptive options. Nevertheless, global sustainability challenges, biodiversity decline, and climate change urgently demand considerable social-ecological transformations towards more ecologically positive futures. Change is continuous, though, and the future will not be like the past.

Hoping to keep everything the same is not an option. Policy overhaul for re-designing the future towards ecological sustainability needs to be thoughtful, creative, interdisciplinary, and probably radically different from the past. Regenerative, sustainable and resilient social-ecological systems need to develop through human interactions and institutions that are future-looking. Science and policy have generally not worked well together to restore (or rehabilitate) complex ecological assemblages, landscape scale, and ecosystem services. Restoration to past conditions in many contexts is either no longer achievable or desirable. New ecological combinations and alternate states might be designed to be more effective in sustaining diverse ecosystem services and future resilience. Ecological restoration must be future-focused, synthesizing social and institutional change to contribute positive future options that become part of whole social-ecological systems, with a redirection and 'restoration' of a nature-positive resilience—that is, transformative, nature-positive change that opens further pathways for resilient futures, especially in urban and peri-urban environments.

Landscape and urban sciences contribute a valuable framework for understanding contextually relevant social-ecological system interactions through which to creatively design and plan for transitions. Social-ecological, nature-positive resilience requires a transformative process that incorporates, assesses, and understands a diverse array of design options and transition routes. Transformative nature-positive SES restoration requires knowledge and understanding of trend trajectories and alternative landscape futures scenarios for a particular SES regional context. Further understanding of the circumstances, timing, and windows of opportunity are also required, while building in options for future adaptation towards more desirable and sustainable (ecologically positive) alternative landscape futures [7].

While the challenges are complex, there are examples of successful, timely transformative adaptation that becomes ongoing, creating SES resilience [3,39]. David Cash and colleagues examined applications of knowledge systems for (ecologically) sustainable development and concluded, "all else being equal, those systems that made a serious commitment to managing boundaries between expertise and decision making, more effectively linked knowledge to action than those that did not" [17] (p. 8089). At the nexus of science and policy, there are increasing calls for 'foresight intelligence' [6,25,34,64]. With

advanced new policy methods that incorporate an understanding of the cycles of complex interdependencies of SES, institutional capacity can be built to identify the properties and characteristics of emerging tipping points to change direction towards more sustainable futures and build capacity for more responsive future adaptation [7]. Enduring sustainability requires transformative responses grounded in the understanding of multiple ALFs, which provide future options redundancy (FOR)—the essence of adaptive capacity. A FOR plan contributes a variety of possible pathways and alternative futures within the characteristics, change pressures, and capacity of a region. A policy-relevant operational model can then be developed, through closely integrated landscape, institutional, and policy sciences, that regularly explores plausible transformations to facilitate a sustainable society. Understanding the characteristics of systems change will help identify 'leverage' points where transformations might be assisted in the future. Such foresight capabilities, of recognizing and understanding in advance the accumulating circumstances creating opportunities for SES restoration, will be the mark of a sustainable society.

**Author Contributions:** Both D.J.B. and E.J.T. contribute equally to conceptualization, methodology, resources, writing—original draft preparation, and writing—review and editing. All authors have read and agreed to the published version of the manuscript.

**Funding:** We thank funding agencies in Australia (ARC, LWA, RIRDC, NCCARF), the USA (NSF, F&WS, USDA) and the EU (FoPER) for supporting allied research.

**Data Availability Statement:** Not applicable.

**Acknowledgments:** Peter Bridgewater, Judith McNeil, Ian Reeve and David Mouat read early manuscript drafts and provided valuable suggestions.

**Conflicts of Interest:** The authors declare no conflict of interest.

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
