# Peer review of "Future Options Redundancy Planning: Designing Multiple Pathways to Resilience in Urban and Landscape Systems Facing Complex Change"

_urbansci, doi:10.3390/urbansci7010011_

Round 1

Reviewer 1 Report

Thank you for having the opportunity to review this paper. It conntains a relevant proposal in the field of urban landscape planning while showing a novel idea of Future Options Redundancy (FOR) as a wide-open idea to rearrange complex socioecological, psychological, spatial, institutional, historical, political aspects os the landscape with the aim to achieve a sustainable transition of landscape arrange and use.

Nevertheless, I feel the lack of clarity on the elements of FOR, what are their pillars, how they caqn be clearly described and inter-related. I feel these elements are disperse along the sections, and they should be explicitly shown to the readers. My recommendation is to provide a subsection that can make it clear.

We already have a well established set of knowledge on comprehensive impact assessmentin the Strategic Environmental Assessment (SEA) - see, for instance, Morrison-Saunders, Bina, Cashmore, Bond etc.  In fact, the SEA thinking could provide answers to integrate policies, plans, and projects with the current develoment initiatives, but we keep creating other surrogates to similar problems. 

Another doubts I have while reading:

- line 14 - What does mean urban and "other development"? (Does it refer to productive activity?)

- line 18 - restore "what was" - The restoration concept refers to recreation, initiation, acceleration of disturbed ecosystem recovery. How can the restoration principles of provision (sustainment), regulation, supporting...) be contextualised in this research?

How does FOR express complexity, connectivity, redundancy?

In the keywors, why is present the expression "nature-positive"?

The Introduction mixes themes from ecology of the ecosystems and environmental public policiies, but this last one are only superficially addressed. |For instance, the "command-control" public policies.

- line 73 - What does mean policy-focused knowledge systems?

- line 76 - What does mean "ecologically negative ways"? Does it suppose not recognising environmental decay, destruction, or neglect it?

- lines 81-82 - What type of human-nature interaction? Here we have a premise according to which is enough the human-nature interaction to restore the environment without qualifying such interactions.

- Figure 1 - What are the differencs between Alternative, Future, Options?

After the Figure 1, the authors discuss redundancy in ecological aspects and socio-spatial aspects as well. However, still remains a gap related to institutional and political redundancy - what it will look like? The authors recognise the need for change in SES research in psychology, sociology, economics, but not in historical dimension.

- lines 337-350 - Please, explain clearly FR in terms of human resources, social institutional options; emerging circumstances: please, explain it in details.

- I think ALF should be developed in a separated subsection. It does not appear earlier in the manuscript.

My overall assessment is that the work is valuable, but it needs further development to clearly show the elements and respective relationships of FOR and how it is different from promoting the SEA.

Author Response

Thank you for your review. Please see out attached responses to all of your comments.

Reviewer 2 Report

Anthropogenic activity in cities affects the state of the environment, changes the ecosystem, accordingly there is a need to develop scenarios for the future state of the environment and develop measures to minimize negative scenarios in particular. This article is devoted to these aspects, in particular to the description of the Future Options Redundancy (FOR) planning methodology for developing the resilience of urban and landscape systems facing complex changes.

The article describes the importance of socio-ecological information in spatial planning, restoration of the urban environment, a model of various scenarios for the development of socio-ecological systems has been developed, taking into account institutional levels, spatial scaling, time frames, the place of Future Options Redundancy (FOR) in this system has been showed. The essence of the concept of Future Options Redundancy (FOR), the relationship between environmental policy and the peculiarities of the flow of environmental processes is characterized.

The article was written at a fairly high scientific and methodological level, but there are a number of comments and questions for the authors:

• The purpose and task of the article is not clear (there is no such section).

• It would be appropriate to graphically summarize the essence of the concept of Future Options Redundancy (FOR) and make a comparison with other similar concepts.

• The article will benefits if the implementation of Future Options Redundancy (FOR) data shows on the example of a specific territory.

• Conclusions should be written in accordance with the objectives of the article.

Author Response

Thank you for your review. Please see the attached response to all of your comments.

Reviewer 3 Report

In the present study, the authors developed a revision presenting some new useful ideas for SES resilience. Authors explained the importance that multiple disciplines and alternative pathways have in ensuring SES integrity. However, they lack in supporting what disciplines and pathways should be encouraged in this sense, and the manuscript is so disorganized, that even if mentioned, the reader will have the doubt in that sense. Besides, as the introduction repeat itself through the vision of multiple disciplines and the need of different pathways, I would first recommend authors to avoid redundancy, which will help to reduce confusion for readers. Moreover, considering the need to maintain essential information, authors should also consider the need to explicitly state the purpose of all sentences, as many of them leave the reader wondering about risen concepts and their context within the revision. I order to increase audience reachability, the manuscript should be comprehensible in all its aspects and for a plethora of readers. Moreover, while the introduction seems to need enhanced, the section 2 is very hard to understand. Much of sentences are overcomplicated, and there is a lack of coherence on their sequencies, because seem that appeared, again, without context. Therefore, the entire section 2 needs to be rewritten in a way that reader will understand why they need to know this kind of information for SES, in a simple, to the point, and clear home-message. The same can be found within the section 3, which have valid sentences if taken separately, but in their entirety, they seem a pile of arguments without direction to the conclusion, that thereby appear as not robust. Such a review that authors proposed seems more like a perspective (or a pile of them), for which I would suggest a re-submission in this sense. However, prior the submission, authors need to clarify and re-organize the entire manuscript. In summary, although with novel perspectives and valid arguments, the worst thing is that the reader has no compass during the reading of the manuscript, which appears like a storm of detached ideas, rather than a flow that escort the reader to new insights. As a reviewer I cannot go further in specific details without waiting for a more organised version.

Specific comments:

L49. Please add a reference for SES

L49-51. This sentence is too vague. Please specify: Greater in what? What outcomes? I cannot understand the full meaning of the sentence.

L51-52. Please add a reference for emergent properties, as this concept may vary and therefore create confusion, especially in cross-disciplinary studies.

L57-59. The review seems to refer to anthropogenic influences causing disruption of natural systems and biodiversity. However, here is referred to natural disasters. This part needs more context to be adequate to this study.

L59-63. The sentence is unclear and too vague, or incorporeal. Please be more specific, even if in a broad sense.

L68-69. This sentence is problematic, as seems that no solution may emerge. Please be more specific on “silver bullets”. Their inadequacy may regard single disciplines studies? If yes it should be explicitly stated. I am assuming that because the authors proposed that solution may derive from interdisciplinary studies.

L75-78. This is a very important sentence. Maybe need a better placement to strengthen the home-message of the manuscript.

L79-81. I know that maybe it is difficult or inadequate to be specific, but this sentence seems too vague and intangible to me.

L83. “However, the future will not be like the past.” This seems more adapted to a “perspective” than a “review” paper.

L85. Caution is needed on “Novel ecological systems”, because it may be misinterpreted.

L88-98. Various examples on SES are indeed necessary to better understand SES context. However, again such examples need more details and insights on how multiple SES are addressed, through multiple disciplines.

L110. Please more details are needed on “foresight intelligence”

L116. Authors need to change the title with one more adapted to the section content. Maybe something like the importance of defining a ‘place’.

L124-129. This is very unnecessary for SES, or it “appeared” with a lack of context.

L131. Meaningful to what? Please detail is needed.

L132-135. Is this information relevant? Why?

L137-143. The concept of defining a place that have historical relevance for SES seems to link with the sentence in abstract: “. Ongoing sustainability requires urban and landscape designs that provide ongoing anticipatory, restorative nature-positive capacity in the context of future change and pressures.”. However, this link seems lacking after reading the entire section. Maybe authors need to be mor explicit in that sense.

L153-161. I figure that depict the SES characteristics would be greatly useful to the reader.

L193-194. This sentence is difficult to understand.

Author Response

Thank you, in particularly, for your review. We have worked to address all of your comments and concerns (please see attached) and feel the manuscript is much better now because of these edits. 
